# A Review of Hydrogen Purification Technologies for Fuel Cell Vehicles

Zhemin Du [1,2,†] , Congmin Liu [2,†], Junxiang Zhai [2], Xiuying Guo [2], Yalin Xiong [3], Wei Su [1,*] and Guangli He [2,*]

[1] Tianjin Key Laboratory of Membrane and Desalination Technology, School of Chemical Engineering and Technology, Tianjin University, Tianjin 300350, China; duzhemin1997@tju.edu.cn
[2] National Institute of Clean-and-Low-Carbon Energy, Future Science City, Changping District, Beijing 102211, China; congmin.liu@chnenergy.com.cn (C.L.); junxiang.zhai.a@chnenergy.com.cn (J.Z.); xiuying.guo.e@chnenergy.com.cn (X.G.)
[3] China Energy Hydrogen Technology, Guohua Investment Building, 3 South Street, Dongcheng District, Beijing 100007, China; yalin.xiong@chnenergy.com.cn
* Correspondence: suweihb@tju.edu.cn (W.S.); guangli.he@chnenergy.com.cn (G.H.); Tel.: +86-022-2740-3389 (W.S.); +86-010-5733-9646 (G.H.)
† These authors contributed equally to this work.

**Abstract:** Nowadays, we face a series of global challenges, including the growing depletion of fossil energy, environmental pollution, and global warming. The replacement of coal, petroleum, and natural gas by secondary energy resources is vital for sustainable development. Hydrogen ($H_2$) energy is considered the ultimate energy in the 21st century because of its diverse sources, cleanliness, low carbon emission, flexibility, and high efficiency. $H_2$ fuel cell vehicles are commonly the end-point application of $H_2$ energy. Owing to their zero carbon emission, they are gradually replacing traditional vehicles powered by fossil fuel. As the $H_2$ fuel cell vehicle industry rapidly develops, $H_2$ fuel supply, especially $H_2$ quality, attracts increasing attention. Compared with $H_2$ for industrial use, the $H_2$ purity requirements for fuel cells are not high. Still, the impurity content is strictly controlled since even a low amount of some impurities may irreversibly damage fuel cells' performance and running life. This paper reviews different versions of current standards concerning $H_2$ for fuel cell vehicles in China and abroad. Furthermore, we analyze the causes and developing trends for the changes in these standards in detail. On the other hand, according to characteristics of $H_2$ for fuel cell vehicles, standard $H_2$ purification technologies, such as pressure swing adsorption (PSA), membrane separation and metal hydride separation, were analyzed, and the latest research progress was reviewed.

**Keywords:** hydrogen energy and fuel cells; impurity; hydrogen purification

## 1. Introduction

Energy resource depletion and global warming are severe challenges of our modern society. The transportation industry plays an essential role in energy consumption and greenhouse gas emissions. According to the International Energy Agency (IEA), it was responsible for 29% of global energy consumption in 2017 and 25% of global carbon dioxide emission in 2016 [1]. Hydrogen ($H_2$) fuel cells provide zero pollutant discharge. The authorities in many countries have strongly supported the production of fuel cell vehicles, and this initiative will inevitably become the future developmental direction in the automotive industry. The USA was the first country that set $H_2$ energy and fuel cells as a long-term energy strategy. There were 5899 fuel cell vehicles in the USA by the end of 2018 [2]. Concerning the promotion of $H_2$ fuel cell vehicles, Japanese and South Korean companies were pioneers in large-scale mass production, successfully launching various mass-produced vehicles, such as Toyota Mirai, Honda Clarity, and Hyundai Nexo [3]. Since then, four automobile group alliances have gradually been formed, including Daimler, Ford, and Renault–Nissan, General Motors and Honda, Bayerische Motoren Werke (BMW)

and Toyota, and Audi and Hyundai. The alliances invested joint effort in developing $H_2$ fuel cell vehicle technologies, and accelerated their commercialization. The Shanghai Automotive Industry Corporation of China launched the fourth fuel cell vehicle using a Roewe 950 vehicle with a 400 km driving range without refueling, demonstrating its capacity for small-scale production [4].

$H_2$ fuel cells mainly include phosphoric acid fuel cells (PAFCs), molten carbonate fuel cells (MCFCs), solid oxide fuel cells (SOFCs), alkaline fuel cells (AFCs), and proton-exchange membrane fuel cells (PEMFCs) [5]. PEMFCs are dominant since they possess a high power density, low-temperature start, and compact structure, representing an ideal power source for $H_2$ fuel cell vehicles. On the other side, PEMFCs require high-purity $H_2$. Otherwise, the fuel cell performance and running life may be severely affected [6]. Currently, $H_2$ production technologies, such as coal gasification, natural gas steam reforming, methanol reforming, and water electrolysis, are very well established in China [7]. According to statistical data from the China Hydrogen Alliance and China National Petroleum and Chemical Planning Institute, the current $H_2$ production capacity in China is approximately 41 million tons/year, with a yield of 33.42 million tons. Specifically, the yield of $H_2$ as an independent component (synthetic gas not containing $H_2$), which meets the quality standards of $H_2$ for industrial use and can be directly sold as industrial gas, is about 12.7 million tons/year. Among these, the $H_2$ yield produced from coal is the highest (21.24 million tons), accounting for 63.54%, followed by $H_2$ produced from by-product gas (7.08 million tons), natural gas (4.6 million tons), and electrolyzed water (0.5 million tons). However, the $H_2$ contributions from supercritical steam coal [8], photocatalytic water decomposition with solar energy [9], and biological $H_2$ production [10] are still in the research and developmental stage (Table 1). Different raw materials yield large differences in the composition and impurity contents of $H_2$ produced using various technologies. Thus, efficient $H_2$ purification technologies that enable the removal of impurities from $H_2$ and provide high-qualify $H_2$ for fuel cell vehicles are of the utmost importance for developing the $H_2$ fuel cell vehicle industry.

**Table 1.** Emerging $H_2$ production methods.

| $H_2$ Production Method | Technical Feature |
| --- | --- |
| $H_2$ production from supercritical steam coal | In this technology, supercritical water [namely, temperature and pressure are at or above the critical values (374.3 °C and 2.1 MPa)] is used as a medium that provides a homogeneous and high-speed reaction because of its special physical and chemical properties, so that the chemical energy of coal is directly and efficiently converted into hydrogen energy [8]. |
| $H_2$ production from water photocatalytically decomposed by solar energy | Photocatalyst powders or electrodes can produce photo-generated carriers by absorbing solar energy, so they decompose water into $H_2$ and $O_2$. The photocatalytic $H_2$ production can be subdivided mainly into heterogeneous photocatalytic (HPC) $H_2$ production and photo-electrochemical (PEC) $H_2$ production [9]. |
| Biological $H_2$ production | $H_2$ is a product of microorganisms' metabolism using biomass and organic wastewater as raw materials. Based on the type of microorganisms and their metabolic mechanisms, the biological $H_2$ production technology includes water splitting $H_2$ production, photo-fermentative $H_2$ production, dark fermentative $H_2$ production, and $H_2$ production combined with photo-fermentation and dark fermentation [10]. |

To support large-scale applications of $H_2$ energy in the transportation field, novel and high-efficient purification technologies for the production of low-cost and high-quality $H_2$ should be urgently developed. In this study, the characteristics of standard $H_2$ purification technologies, such as pressure swing adsorption (PSA), membrane separation and metal hydride separation, were analyzed. Research progress was reviewed according to the characteristics of $H_2$ for fuel cell vehicles. To further improve the separation efficiency, it is necessary to continuously conduct studies on the novel and highly selective adsorption materials, long-lasting and low-cost membrane materials, anti-poisoning metal hydride materials with a low regeneration energy consumption, as well as new separation and coupling processes based on the materials mentioned above.

## 2. $H_2$ Standards for Fuel Cell Vehicles

The International Organization for Standardization (ISO) issued the ISO 14687-2:2012 standard in 2012, and the Society of Automotive Engineers (SAE) issued the SAE J2719-201511 standard in 2015, presenting the same requirements for $H_2$ quality for PEMFCs. Until 2019, China was following the GB/T 3634.2-2011 Hydrogen Part 2: Pure Hydrogen, High-Pure Hydrogen, and Ultrapure Hydrogen standard. However, this standard was aimed at industrial $H_2$ use, limiting the impurity content only partly, without specific regulations on other impurities that may affect $H_2$ fuel cells' performance. Therefore, by the end of 2018, China set the GB/T 37244-2018 Fuel Specification for Proton Exchange Membrane Fuel Cell Vehicles—Hydrogen standard, which was in line with ISO 14687-2:2012 and SAE J2719-201511 standards, regulating the concentration of fourteen impurities: water ($H_2O$), total hydrocarbon (HC) (by methane), oxygen ($O_2$), helium (He), nitrogen ($N_2$), argon (Ar), carbon dioxide ($CO_2$), carbon monoxide (CO), total sulfide (by $H_2S$), formaldehyde (HCHO), formic acid (HCOOH), ammonia ($NH_3$), total halide (by halide ions), and maximum particulate matter. The PEMFC technology was remarkably improved, e.g., a lower Pt usage, thinner electrolyte membrane, and higher operating electric current density and lower humidity, so it is necessary to reconsider the previously set impurity limit in $H_2$. The ISO technical committee for $H_2$ energy, IOS/TC 197, issued the ISO 14687:2019 standard in November 2019, combining and revising three $H_2$ fuel cell-related standards, namely, ISO 14687-1, ISO 14687-2, and ISO 14687-3. Meanwhile, according to the ISO 14687:2019 standard, the SAE issued the SAE J2719-202003 standard in March 2020, extending the limit of $CH_4$, $N_2$, Ar, and HCHO impurities. Table 2 shows the requirements for the impurity content in $H_2$ for fuel cells, including previous and new standards in China and abroad.

**Table 2.** Requirements for the impurity content in $H_2$ for fuel cells in previous and new standards in China and abroad.

| Component | GB/T 3634.2-2011 | | | ISO 14687-2:2012 SAE J2719-201511 GB/T 37244-2018 | ISO 14687:2019 SAE J2719-202003 |
|---|---|---|---|---|---|
| | Pure $H_2$ | High Pure $H_2$ | Ultrapure $H_2$ | | |
| $H_2$ purity (mole fraction) | 99.99% | 99.999% | 99.9999% | 99.97% | 99.97% |
| Total non-hydrogen gases | - | 10 ppm | 1 ppm | 300 ppm | 300 ppm |
| $H_2O$ | 10 ppm | 3 ppm | 0.5 ppm | 5 ppm | 5 ppm |
| Total HC (by methane) | - | - | - | 2 ppm | - |
| Non-methane HC (by $C_1$) | - | - | - | - | 2 ppm |
| Methane | 10 ppm | 1 ppm | 0.2 ppm | - | 100 ppm |
| $O_2$ | 5 ppm | 1 ppm | 0.2 ppm | 5 ppm | 5 ppm |
| He | - | - | - | 300 ppm | 300 ppm |
| $N_2$ and Ar | - | - | - | 100 ppm | - |
| $N_2$ | 60 ppm | 5 ppm | 0.4 ppm | - | 300 ppm |
| Ar | Agreed by supply and demand | Agreed by supply and demand | 0.2 ppm | - | 300 ppm |
| $CO_2$ | 5 ppm | 1 ppm | 0.1 ppm | 2 ppm | 2 ppm |
| CO | 5 ppm | 1 ppm | 0.1 ppm | 0.2 ppm | 0.2 ppm |

**Table 2.** *Cont.*

| Component | GB/T 3634.2-2011 | | | ISO 14687-2:2012 SAE J2719-201511 GB/T 37244-2018 | ISO 14687:2019 SAE J2719-202003 |
|---|---|---|---|---|---|
| | Pure $H_2$ | High Pure $H_2$ | Ultrapure $H_2$ | | |
| Total sulfide (by $H_2S$) | - | - | - | 0.004 ppm | 0.004 ppm |
| HCHO | - | - | - | 0.01 ppm | 0.2 ppm |
| HCOOH | - | - | - | 0.2 ppm | 0.2 ppm |
| $NH_3$ | - | - | - | 0.1 ppm | 0.1 ppm |
| Total halide (by halide ion) | - | - | - | 0.05 ppm | 0.05 ppm |
| The concentration of maximum particulate matter | - | - | - | 1 mg/kg | 1 mg/kg |

## 3. The Impact of Impurities on Fuel Cells

As shown in Table 1, compared with $H_2$ for industrial applications, the requirements for $H_2$ purity for fuel cells are not high. Still, the impurity content is strictly controlled, determined by fuel cells' structure and operating characteristics. For instance, even a low CO content may cause irreversible damage to the performance and running life of fuel cells. Table 3 shows the impact of excessive impurities on fuel cells.

**Table 3.** Impact of impurities on the performance of fuel cells.

| Impurity | Damage Induced by Excessive Impurities |
|---|---|
| $H_2O$ | $H_2O$ can transport water-soluble impurities, such as $Na^+$ and $K^+$, and reduce the membrane proton conductivity. Excessive $H_2O$ induced corrosion of metal parts [11]. |
| HC | Most HCs adsorbed onto the catalyst layer will decrease catalytic performance. Methane does not pollute fuel cells, but it dilutes $H_2$ and hampers performance [12]. |
| $O_2$ | $O_2$ in specific concentrations negatively affects the performance of metal hydride, a type of $H_2$ storage material [11]. |
| Inert gas | Dilution and diffusion of He, Ar, and $N_2$ in $H_2$ decrease the electric potential of fuel cells [13]. |
| $CO_2$ | $CO_2$ has a dilution effect on $H_2$. $CO_2$ in high concentrations can be converted into CO through a reverse water gas shift reaction, thereby leading to catalyst poisoning [14]. |
| CO | CO closely binds to the active site of Pt catalysts, decreasing the effective electrochemical surface area used for $H_2$ adsorption and oxidation [15]. |
| Sulfide | The adsorption of sulfides on the active catalyst sites prevents $H_2$ adsorption on the catalyst surface. The adsorbed sulfides react with Pt catalysts to form stable Pt sulfides, irreversibly degrading the fuel cell performance [16]. |
| HCHO and HCOOH | HCHO and HCOOH are adsorbed on catalysts to form CO, thereby leading to catalyst poisoning [17]. |
| $NH_3$ | $NH_4^+$ can reduce the proton conductivity of the ionic polymer. $NH_3$ adsorbed on the surface of the catalyst blocks the active sites [18]. |
| Halide | Halide adsorbed on the catalyst layer decreases the superficial area of catalysts. Chloride ions are deposited in the fuel cell membrane by forming soluble chlorides, leading to the Pt catalyst's dissolution [19]. |
| Particulate matter | Particulate matters adsorbed on the active site of catalysts of fuel cells prevent the $H_2$ adsorption on the catalyst surface, blocking the filter and destroying the full cell components [20]. |

## 4. $H_2$ Purification Technology

$H_2$ purification technology is a crucial link from $H_2$ production to $H_2$ utilization. Stable, reliable, and low-cost $H_2$ sources represent a base for large-scale applications of fuel cell vehicles. Thus, high-efficient and low-power $H_2$ purification technologies for fuel cell vehicles play an underlying role in the development of the $H_2$ energy industry.

A fuel cell power system can operate efficiently only if high-quality $H_2$ is provided. $H_2$ produced in coal gasification, natural gas reforming, by-product $H_2$, or from water electrolysis, is collectively referred to as crude hydrogen. It cannot be directly used for fuel cell vehicles without purification according to the existing standards. The composition of different types of crude $H_2$ is listed in Table 4. The $H_2$ purification methods can be mainly classified as physical and chemical methods [21]. The former include adsorption methods [PSA, temperature swing adsorption (TSA), and vacuum adsorption], low-temperature separation methods (cryogenic distillation and low-temperature adsorption), and membrane separation methods (inorganic membrane and organic membrane), while the latter involve a metal hydride separation and catalysis method (Figure 1). The selection of an appropriate $H_2$ purification method is closely related to the hydrogen supply mode and gas source. For $H_2$ production by centralized large-scale coal gasification and natural gas reforming with an $H_2$ supply amount $\geq$10,000 $Nm^3$/h, PSA purification is primarily adopted after transformation, desulfurization, and decarbonization. The PSA technology has been around for a while, and is characterized by low operation costs and a long service life. However, the $H_2$ for fuel cell vehicles produced via traditional PSA with a standard impurity content results in a decreased recovery rate and yield. It is also not cost-efficient due to low requirements for specific impurity removal (e.g., CO $\leq$ 0.2 ppm). Cryogenic distillation is also applicable to large-scale production, but standard $H_2$ purity is 85–99%, which does not satisfy the application requirements. For $H_2$ production by centralized by-product mode with an $H_2$ supply of 1000–10,000 $Nm^3$/h, versatile processes should be applied based on different impurities to improve the $H_2$ recovery efficiency. For example, an organic membrane combined with a PSA process is used for obtaining methanol purge gas, while a two-stage or multi-stage PSA process is adopted for obtaining coke oven gas and by-product gas from the refinery. Concerning such small-scale on-site distributed $H_2$ production scenarios, with the $H_2$ supply $\leq$ 1000 $Nm^3$/h and vehicle $H_2$ supply, traditional PSA separation shows the disadvantages of large floor area, inflexibility, and low adaptability. Hence, low-temperature adsorption, metal hydride, and metal membrane separations are available processes according to the types and amounts of impurities. Low-temperature adsorption can effectively eliminate multiple impurities, such as sulfide, HCHO, and HCOOH. However, it requires high energy consumption, and it is a complex process suitable for special small-scale and cold source applications [22]. Metal hydride separation and palladium (Pd) membrane separation methods are reasonably effective in separating gas sources with a high content of inert components. At the same time, their inherent disadvantage is that purified materials react with impure gas during the $H_2$ recovery, reducing the purification efficiency [23]. New membrane technologies, such as carbon molecular sieve membranes (CMSMs) [24], ionic liquid membranes [25], and electrochemical $H_2$ pump membranes [26], are currently hot spots in scientific research. However, their industrial-scale implementation is still hard to foresee.

**Table 4.** Composition of different types of crude hydrogen.

| Component (%) | $H_2$ | CO | $CO_2$ | $CH_4$ | $N_2$ | Ar | Total Sulfur | $H_2O$ | $O_2$ | Others |
|---|---|---|---|---|---|---|---|---|---|---|
| Coal gasification [27] | 25–35 | 35–45 | 15–25 | 0.1–0.3 | 0.5–1 | - | 0.2–1 | 15–20 | - | - |
| Natural gas reforming [28] | 70–75 | 10–15 | 10–15 | 1–3 | 0.1–0.5 | - | - | - | - | - |
| Methanol reforming [29] | 75–80 | 0.5–2 | 20–25 | - | - | - | - | - | - | - |
| Coke oven gas [30] | 45–60 | 5–10 | 2–5 | 25–30 | 2–5 | - | 0.01–0.5 | - | 0.2–0.5 | 2–5 |
| Methanol purge gas [31] | 70–80 | 4–8 | 5–10 | 2–8 | 5–15 | 0.1–2 | - | - | - | - |
| Synthetic ammonia tail gas [32] | 60–75 | - | - | - | 15–20 | - | - | 1–3 | 10–15 | - |
| Biomass gasification [33] | 25–35 | 30–40 | 10–15 | 10–20 | 1 | - | 0.2–1 | - | 0.3–1 | - |

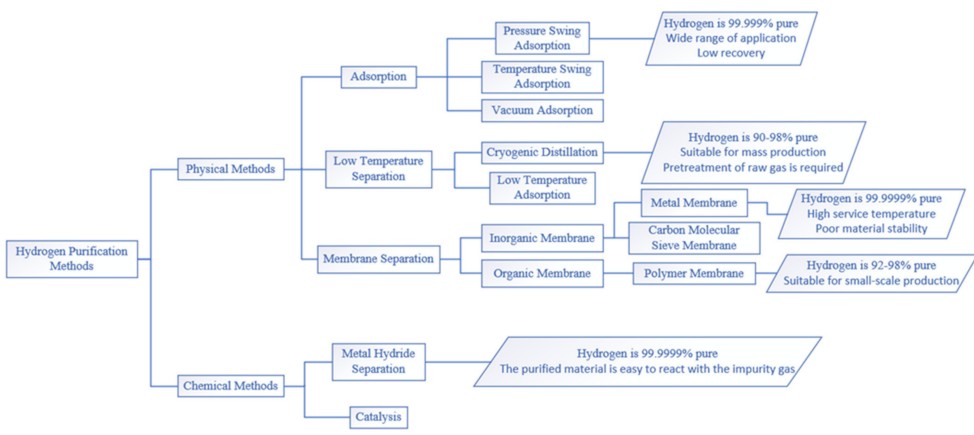

**Figure 1.** Classification of hydrogen purification technologies.

### 4.1. PSA Methods

Gas separation and purification by PSA are implemented by periodical pressure changing based on the difference in the adsorbent capacity for different gases. The PSA separation effect primarily depends on the type of adsorbent and the technical process used. $H_2$ significantly differs from the majority of gas molecules, such as $CO_2$, CO, and $CH_4$, in terms of static capacity, so it is very suitable for PSA separation and purification [34]. Air Product, Air Liquid, and other renowned global gas companies, are already established and very successful examples for the industrial application of $H_2$.

Traditional adsorbents include zeolite molecular sieves, activated carbon, activated alumina, and silica gel. Modifications and innovations of these adsorbents were reported in regard to different impurities, and most of the studies focused on $CO_2$ removal. Lively et al. [35] used hollow fibers as an adsorbent to investigate the $CO_2$ removal in PSA experimental devices. The purity of the obtained $H_2$ was 99.2% pure, with a recovery rate of 88.1%, and this needs to be optimized. Shamsudin et al. [36] increased the $H_2$ purity to about 100% and the recovery rate to 88.43% via the strong $CO_2$ absorption of palm shell charcoal. He et al. [37] reported a structured activated carbon system applied to rapid PSA (RPSA) using a dip-coated Ni foam framework. Under the working conditions of 0.4 MPa and 200 mL/min, the adsorption rate constant K was 0.0029 $s^{-1}$, which was about two times higher than that of traditional adsorbents. The material exhibited a better $CO_2$ adsorption effect in $H_2$. Moreover, Kuroda et al. [38] applied hydroxyl aluminum silicate clay (HAS-Clay) to purify the $H_2$ produced by biomass, and found a relatively high adsorption selectivity to $CO_2$. This adsorbent is also applicable for the adsorption and separation of $H_2S$. Metal-organic frameworks (MOFs) are recently developed materials with easily adjustable structures and properties, and they are ideal novel adsorption materials. Agueda et al. [39] utilized UTSA-16 as an adsorbent to remove the $CO_2$ impurities, and simulated the PSA process of the steam methane reforming of tail gas. The results revealed a $H_2$ purity up to 99.99–99.999%, a recovery rate of 93–96%, and a yield of 2–2.8 mol/kg/h.

Researchers are trying to develop novel adsorbents for the simultaneous removal of multiple impurities in $H_2$. Brea et al. [40] synthesized a raw material NaX molecular sieve within CaX and MgX molecular sieves using an ion-exchange method. They conducted an adsorption simulation for the $H_2/CH_4/CO/CO_2$ gas mixture and showed that these three adsorbents could yield $H_2$ with a purity higher than 99.99%. The CaX molecular sieve application exhibited the highest recovery rate and yield of $H_2$. Besides this, Banu et al. [41] compared the performances of four kinds of MOF adsorbents (UiO-66(Zr), UiO-66(Zr)-Br, UiO-67(Zr), and Zr-Cl$_2$AzoBDC) and discovered that UiO-66(Zr)-Br had the best purification effect on $H_2$ produced via methane steam reforming. Relvas et al. [42] prepared a novel Cu-AC-2 adsorbent to process the $H_2/CH_4/CO/CO_2$ gas mixture. The $H_2$ purity exceeded 99.97%, while the CO content declined to 0.17 ppm, reaching the $H_2$ standards required for fuel cell vehicles.

The improvement and optimization of the PSA process are crucial ways to increase the $H_2$ purification efficiency. The flow scheme of the classical PSA system is shown in Figure 2. Ahn et al. [43] used a two-bed PSA and a four-bed PSA to recover $H_2$ from coal gas with $N_2$ as a major impurity. The four-bed PSA process' performance was superior to that of the two-bed PSA process, yielding a $H_2$ purity of 96–99.5% and a recovery rate of 71–85%. Abdeljaoued et al. [44] established a four-bed PSA theoretical model and performed a twelve-step four-bed PSA experiment at room temperature. They investigated the removal of impurities from $H_2$ produced by ethanol steam reforming for fuel cell vehicles. Further optimization was expected to increase the $H_2$ recovery rate above 75%, providing a CO concentration lower than 20 ppm. Moon et al. [45] investigated the eight-bed PSA process. Considering that a $H_2$ purity of 99.99%, the highest $H_2$ recovery rate was 89.7%, which was approximately 11% higher than that of a four-bed PSA process. Moreover, Zhang et al. [46] established the five-step one-bed and six-step two-bed PSA cycle models, and compared them in terms of purity, recovery rate, and yield of $H_2$. The recovery rate and the yield of the two-bed PSA process were 11% and 1 mol/kg/h, respectively, higher than those of the one-bed PSA process. Li et al. [47] explored the effects of adsorption pressure, adsorption time, and P/F ratio on PSA. They designed a six-step two-bed PSA process for the purification of $H_2$ produced by methane steam reforming, and the produced $H_2$ exhibited a purity and a rate of more than 99.95% and 80%, respectively. When the $CH_4$ concentration in impurities was high, it was necessary to increase the adsorption pressure to ensure the purity of the $H_2$. Yáñez et al. [32] developed a four-bed PSA device with a 5 Å molecular sieve as an adsorbent for purification of $H_2$ from synthetic ammonia tail gas ($H_2$:$N_2$:$CH_4$:Ar = 58:25:15:2). It was indicated that the $H_2$ purity was up to 99.25–99.97%, while the recovery rate was 55.5–75.3%.

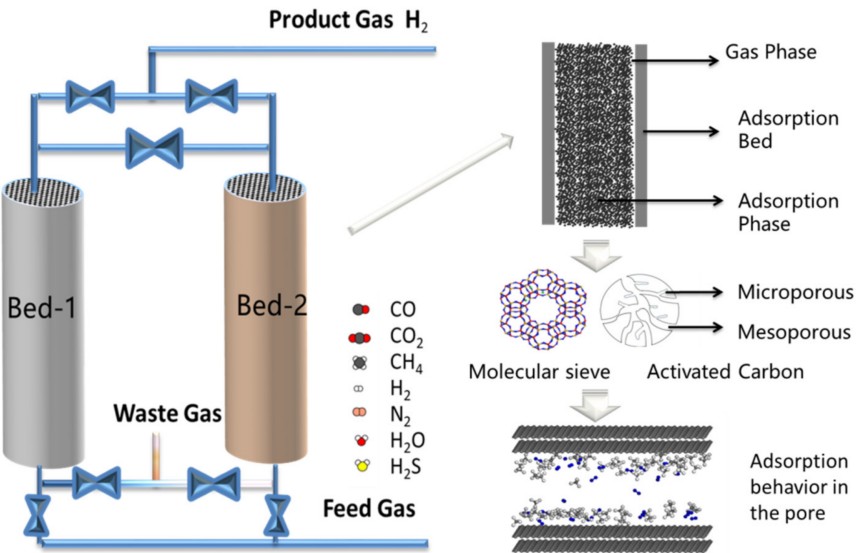

**Figure 2.** Flow scheme of the classical pressure swing adsorption (PSA) system.

The improvement of process flow based on conventional PSA is an important research direction for increasing the $H_2$ purity and recovery rate. Vacuum PSA (VPSA) can forcibly desorb impurities with a strong adsorption capacity from adsorbents via vacuum pumping in order to regenerate adsorbents. You et al. [48] showed that the VPSA and PSA could produce $H_2$ with a similar purity level under the same conditions, but the recovery rate was increased by about 10% during VPSA. Besides this, Lopes et al. [49] performed an experiment on rapid VPSA (RVPSA), and the results illustrated that RVPSA was able to improve the $H_2$ yield by nearly 410% compared with PSA. Golmakani et al. [50] conducted a comparative study on PSA, VPSA, and TSA processes, and discovered that the VPSA process could yield the $H_2$ for fuel cell vehicles with a reasonable cost-efficiency and

recovery rate, proposing it as the best option among the three investigated processes. Furthermore, Golmakani et al. [51] established a sixteen-step four-bed VPSA model. They studied the influence of $N_2$ on the process' performance so as to enhance the recovery rate of $H_2$ produced by VPSA and decrease the energy consumption.

Many novel PSA devices are currently in the development or implementation phase for different impurities and $H_2$ utilization demands. Thus, Air Products [52] has investigated and developed a novel PSA device, named Sour PSA, to capture the acidic impurities, such as $CO_2$ and sulfides, in $H_2$. Majlan et al. [53] designed a compact PSA (CPSA) system with a rapid circulation rate to provide $H_2$ continuously without the need for adsorbent regeneration. Moreover, it could reduce CO concentration in the $H_2/CO/CO_2$ gas mixture from 4000 to 1.4 ppm, and the $CO_2$ concentration from 5% to 7 ppm, yielding $H_2$ with a purity of 99.999%. Zhu et al. [54] proposed a seven-step two-bed elevated-temperature PSA (ET-PSA) system to separate the feed gas $CO/CO_2/H_2O/H_2$ under proper conditions. In this way, $H_2$ with a purity of 99.9991% and a recovery rate of 99.6% was obtained. As such, the system was applicable to the removal of CO and $CO_2$ from $H_2$. A summary of the PSA technology is shown in Table 5.

**Table 5.** Summary of PSA technology.

| Adsorbent | Process Flow | Feed Gas | $H_2$ Purity | $H_2$ Recovery Rate | Reference |
|---|---|---|---|---|---|
| Hollow fiber sorbent | RCPSA | $CO_2$:$H_2$ = 25:75 | 99.2% | 88.1% | [34] |
| Palm kernel shell activated carbon | Two-column PSA | $CO_2$:$H_2$ = 15:85 | About 100% | 88.43% | [35] |
| UTSA-16 | Four-column PSA | Steam methane reforming off-gas | 99.99–99.999% | 93–96% | [38] |
| CaX zeolite | Four-column PSA | $H_2$:$CH_4$:CO:$CO_2$ = 75.89:4.01:3.03:17.07 | +99.99% | 69.6% | [39] |
| Cu-AC-2 | Four-column PSA | $H_2$:$CO_2$:$CH_4$:CO = 70:25:4:1 | +99.97% | +75% | [41] |
| Activated carbon/zeolite 5A | Four-bed PSA | $H_2$:$CO_2$:$CH_4$:CO:$N_2$ = 38:50:1:1:10 | 96–99.5% | 71–85% | [42] |
| Activated carbon | Twelve-step four-column PSA | CO:$CO_2$:$CH_4$:$H_2$ = 1:25:5:69 | 99.999% | +75% | [43] |
| Activated carbon/zeolite LiX | Eight-layered bed PSA | $H_2$:$CO_2$:CO:$N_2$:Ar = 88.75:2.12:2.66:5.44:1.03 | 99.99% | 89.7% | [44] |
| Activated carbon/zeolite 5A | Six-step layered two-bed PSA | $H_2$:$CH_4$:CO:$CO_2$ = 72.9:3.6:4.5:19 | +99.95% | +80% | [46] |
| 5A zeolite | Four-column PSA | $H_2$:$N_2$:$CH_4$:Ar = 58:25:15:2 | 99.25–99.97% | 55.5–75.3% | [32] |
| Activated carbon | CPSA | $H_2$:CO:$CO_2$ = 94.6:0.4:5 | 99.999% | - | [52] |
| Potassium-promoted layered doubleoxide | Two-column seven-step ET-PSA | CO:$CO_2$:$H_2O$:$H_2$ = 1:1:10:88 | 99.9991% | 99.6% | [53] |

### 4.2. Membrane Separation Methods

As an emerging gas separation technology, membrane separation has the advantages of flexible and simple operation, compact structure, low energy consumption, and environmental friendliness. In the membrane separation technology with a perm-selective membrane as a separation medium, the raw material components can selectively permeate the membrane under the action of driving forces (pressure difference, concentration difference, and potential difference), thereby achieving separation and purification [55]. The performance of membrane materials is the most critical factor determining the $H_2$ separation and purification effects of the membrane. Commonly used membrane materials primarily include metal and polymer membranes, and novel membrane materials, such as nanomaterial membrane, CMSM, and MOF membranes, may exhibit preferable separation performance. Therefore, the performance of these membrane materials in the $H_2$ purification is analyzed and evaluated below.

### 4.2.1. Metal Membranes

$H_2$ is catalyzed to protons and electrons on the compact structure of metal membranes. The protons pass through the metal membrane and bind electrons on the other side to form $H_2$ again. However, the metal membrane blocks $CO_2$, $N_2$, $CH_4$, and $O_2$ gas molecules, thereby achieving the selective permeation of $H_2$. Pd membranes are currently the most commonly used metal membranes, due to their excellent $H_2$ permeability, and their high resistance to $H_2$ fluidity and auto-catalytic hydrogenolysis reactions [56]. However, the Pd membrane is related to high manufacturing costs, and it is prone to $H_2$ embrittlement at a low temperature.

A Pd alloy membrane can be formed by adding other metal elements (Ag, Au, Cu, Ni, Y, etc.) into the Pd membrane to solve the $H_2$ embrittlement problem, enlarge the Pd lattice, and increase the $H_2$ permeation rate at the same time. Nayebossadri et al. [57] studied the performance of $H_2$ in natural gas separated by Pd, $PdCu_{53}$, and $PdAg_{24}$ membrane materials at different concentrations. They found that the $H_2$ permeability of the $PdAg_{24}$ membrane is better than that of the other two membranes. Zhao et al. [58] prepared a bilayer bcc–PdCu alloy membrane by the alternative electrodeposition of Pd and Cu on a ceramic support membrane. The membrane exhibited excellent low-temperature tolerance and $H_2$ permeability, and it is a candidate membrane material for $H_2$ separation at ambient temperature.

Both pure Pd membranes and Pd alloy membranes are self-supporting membranes. Their thickness is limited from several tens to several hundreds of micrometers to assure sufficient mechanical strength. When the membrane thickness is too high, it increases the total cost and lowers the $H_2$ permeation rate. As such, it is possible to deposit a Pd membrane or a Pd alloy membrane on the surface of a porous material to prepare a supported Pd composite membrane. The support increases the mechanical strength of the Pd membrane and decreases the Pd amount and membrane thickness, which is beneficial to the total cost and the $H_2$ permeation rate, as described above. Kong et al. [59] deposited a nanoscale Pd membrane on polybenzimidazole-4,4′-(hexafluoroisopropylidene)-bis(benzoic acid) (PBI-HFA) using the vacuum electroless plating (VELP) technique. The novel Pd/PBI-HFA composite membrane completely prevented CO permeation and exhibited good $H_2/N_2$ and $H_2/CO_2$ selectivity. Kiadehi et al. [60] deposited a NaY molecular sieve and a Pd membrane on porous stainless-steel substrates. The permeation of the $H_2$ and $N_2$ mixture into the prepared Pd/NaY/PSS composite membrane was tested, showing that the membrane's $H_2/N_2$ selectivity was 736 at 450 °C. Moreover, Iulianelli et al. [61] prepared a supported $Pd_{70}$-$Cu_{30}/\gamma$-$Al_2O_3$ thin membrane using the metal vapor synthesis method. The membrane showed $H_2/N_2$ and $H_2/CO_2$ selectivity of 1800 and 6500, respectively, at 400 °C and 50 kPa. Huang et al. [62] used natural mineral Nontronite-15A as a surface coating material of porous $Al_2O_3$ to prepare a Pd/Nontronite-15A/$Al_2O_3$ membrane, lowering the production cost compared to other composite membranes that provide high $H_2$ permeability.

The permeation ability of Pd $H_2$ is not the strongest among metals. It has been recently indicated that vanadium group metals, V, Nb, and Ta, have different bcc lattice structures, higher $H_2$ permeability and mechanical strength, and weaker $H_2$ dissociation and adsorption ability than Pd [63]. However, a compact oxide layer forms on the surface, preventing $H_2$ permeation. As a result, the $H_2$ permeation rate of the thin membrane is not very high, although vanadium group metals have a strong lattice $H_2$ permeation ability. Besides this, these metals are more susceptive to $H_2$ embrittlement than Pd. A useful approach toward this problem was depositing an extremely thin Pd layer plated on both sides of the vanadium group metals to form symmetric composite membranes. In that way, the $H_2$ adsorption and dissociation ability of the Pd membrane was combined with the $H_2$ permeation ability of the vanadium group metals, lowering the total cost. Dolan et al. [64] prepared a Pd-coated vanadium membrane with a tubular structure, and this revealed a high $H_2$ permeability and stability, suitable for $H_2$ separation for fuel cell vehicles. Fasolin et al. [65] applied high-power pulse magnetron sputtering technology to

prepare a Pd/V$_{93}$Pd$_7$/Pd multilayer membrane with a total thickness less than 7 μm on the surface of porous alumina. Besides this, research studies have demonstrated that such V-based thin membranes have similar permeability and higher resistance to H$_2$ embrittlement than Pd-based membranes. Alimov et al. [66] prepared a thin-walled seamless tubular membrane using V–Pd and V–Fe alloys. Furthermore, they manufactured a membrane module by welding 18 membranes, which was applied to extract ultra-pure H$_2$. Jo et al. [67] adopted a Pd/Ta composite membrane for ammonia dehydrogenation, overcoming H$_2$ embrittlement and producing H$_2$ with a purity over 99.9999%, while the NH$_3$ concentration was reduced below 1 ppm. Additionally, they applied the Pd/Ta composite membrane to purify H$_2$ from the CO, CO$_2$, H$_2$O, and H$_2$ gas mixture, yielding a H$_2$ purity of more than 99.999% and a CO content of 10 ppm [68]. Budhi et al. [69] investigated the separation of H$_2$ from the H$_2$ and N$_2$ mixture using a Pd/α-Al$_2$O$_3$ membrane. They achieved a higher H$_2$ recovery rate by adjusting the feed gas flow rate to make the membrane operate under non-steady-state conditions.

### 4.2.2. Polymer Membranes

The working principle of polymer membrane separation is based on the different permeation rates of gases through the polymer membrane. Nowadays, polysulfone (PSF), polyimide (PI), and polyamide are commonly used as polymer membrane materials [70]. An ideal polymer membrane material should possess high selectivity, permeability, thermal stability, and good mechanical performance. However, as a rule of thumb, a highly permeable polymer membrane has low selectivity, and vice versa [71]. Since the trade-off between selectivity and permeability limits the use of polymer membranes, researchers attempted to prepare mixed matrix membranes (MMMs) by adding zeolite, silicon dioxide, CMS, and other inorganic materials into the polymer to improve the overall performance [72]. Rezakazemi et al. [73] added 4A zeolite nanoparticles into a polydimethylsiloxane (PDMS) substrate to prepare the PDMS/4A MMMs. The prepared MMMs experimentally exhibited higher H$_2$/CH$_4$ selectivity and H$_2$ permeability than the pure PDMS membrane. Peydayesh et al. [74] introduced Deca-dodecasil 3R (DDR) zeolite into a Matrimid® 5218 PI substrate to prepare the Matrimid® 5218-DDR MMM, yielding H$_2$ permeability and H$_2$/CH$_4$ that were increased by 100 and 189%, respectively.

In addition, polymer blending could also improve the performance of polymer membranes. Hamid et al. [75] synthesized a PSF/PI membrane that possessed higher H$_2$ permeability and H$_2$/CO$_2$ selectivity (4.4) than a single PSF or PI membrane, with a H$_2$ purification efficiency of 80%. Meanwhile, the PSF/PI membrane exhibited more stable physical and chemical properties, yielding a novel polymer membrane with excellent performance. Structurally, the mechanical performance and specific surface area of a hollow fiber membrane are superior to those of a traditional plate membrane. These findings are also used as a developmental direction of the gas separation membrane. Naderi et al. [76] developed a bilayer hollow fiber membrane with a polybenzimidazole (PBI) and sulfonated polyphenylenesulfone (sPPSU) mixture as an outer selection layer, and PSF as an inner support layer. The experimental results indicated a H$_2$ permeability in the membrane of 16.7 GPU, and a H$_2$/CO$_2$ selectivity of 9.7 at 90 °C and 14 atm. Therefore, the membrane was suitable for H$_2$ and CO$_2$ separation at high temperatures.

All the polymer membranes mentioned above have very high H$_2$ selectivity. Furthermore, researchers have developed separation membranes with CO$_2$ selectivity to remove CO$_2$ from H$_2$ efficiently. Figure 3 schematically illustrates two selective membranes. Compared with the H$_2$-selective membrane, the CO$_2$-selective membrane requires a smaller area during separation and generates H$_2$ as a product in a high-pressure state, significantly reducing the mechanical energy loss [77]. As the CO$_2$ molecular diameter is larger than that of H$_2$, the polymer membranes should have a particular CO$_2$ affinity to achieve negative selectivity [55]. Abedini et al. [78] prepared a poly(4-methyl-1-pentene) (PMP)/MIL 53 (Al) MMMs membrane by adding MIL 53(Al) MOF into a poly(4-methyl-1-pentene) (PMP) substrate. It was experimentally indicated that the MMMs possessed higher CO$_2$/H$_2$ negative

selectivity and thermal stability than pure PMP membranes. In the meantime, the negative selectivity of MMMs was enhanced with the increase in feed pressure, and it was capable of overcoming the Robeson upper limit. Cao et al. [79] introduced a covalent organic framework (COF) into polyvinyl amine to prepare the PVAM/COF MMMs, with a $CO_2/H_2$ selectivity of 15 and a $CO_2$ permeation rate of 396 GPU. Moreover, Salim et al. [80] prepared novel oxidatively stable membranes containing quaternary ammonium hydroxide, fluoride, and tetrafluoroborate using a crosslinked polyvinyl alcohol–polysiloxane substrate, with a $CO_2$ permeation rate of 100 GPU and $CO_2/H_2$ selectivity greater than 100, and such membranes were expected to be applied to purify $H_2$ for fuel cell vehicles. Nigiz et al. [81] added graphene oxide (GO) into PDMS to prepare nanocomposite membranes, increasing the $CO_2$ permeation rate and $CO_2/H_2$ selectivity. At a GO content of 0.5% and transmembrane pressure of 0.2 MPa, the $CO_2/H_2$ selectivity rose from 7.1 to 11.7, and the $CO_2$ permeability reached 3670 Barrer. Besides this, Chen et al. [77] prepared ZIF-8-TA nanoparticles using a hydrophilic modification of ZIF-8 with tannic acid (TA). The nanoparticles were introduced into a hydrophilic polyvinyl amine substrate to obtain an MMM. Under the feed pressure of 0.12 MPa, the $CO_2$ permeability and $CO_2/H_2$ selectivity were 987 GPU and 31, respectively, providing a preferable $CO_2/H_2$ separation performance.

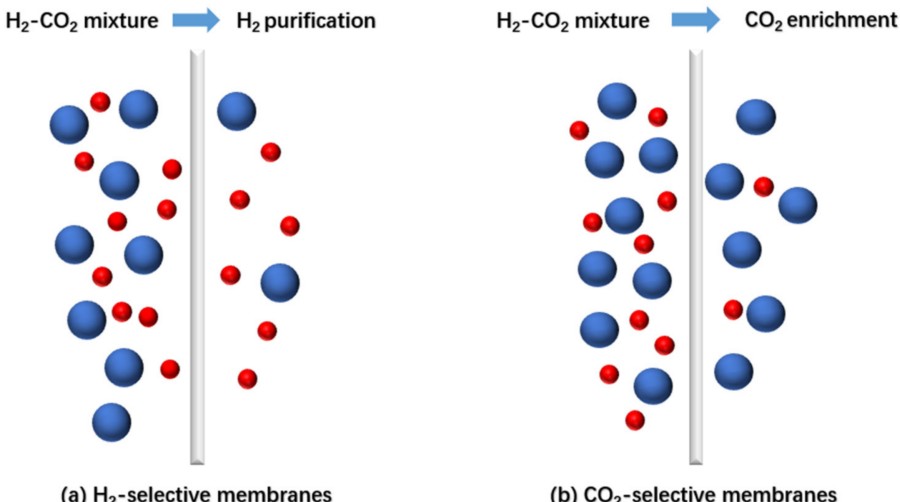

**Figure 3.** Schematic representation of (**a**) $H_2$-selective and (**b**) $CO_2$-selective membranes.

4.2.3. Carbon-Based Membranes

Carbon molecular sieve (CMS) membranes (CMSMs) with an amorphous microporous structure are the most common carbon-based membranes, usually obtained by the carbonization or pyrolysis of polymer precursors in the inert gas or vacuum environment. Common polymer precursors include polyimide and its derivatives, polyfurfuryl alcohols, and phenolic resins [55]. An in-depth exploration was conducted to improve the permeability and selectivity of CMS membranes. Tanco et al. [82] prepared composite alumina–CMS membranes (Al-CMSMs) with tubular porous alumina as a carrier, achieving a $H_2$ and $CH_4$ separation performance considerably better than the Robeson upper limit for polymer membranes at 30 °C. The $H_2$ extracted from an $H_2/CH_4$ gas mixture possessed a purity of 99.4%. Xu et al. [83] prepared CMSMs with ultra-high selectivity by decomposing polyetherketone–cardo polymers at a high temperature. The reported permeability of CMS membranes prepared by carbonization at 700 °C was 5260 Barrer, while the $H_2/CH_4$, $H_2/N_2$, and $H_2/CO$ selectivity was 311, 142, and 75, respectively. When the carbonization was performed at 900 °C, the $H_2/CH_4$ selectivity reached 1859.

Graphene-based membranes, as a new type of carbon-based membranes, have attracted extensive attention in the gas separation field. Graphene and GO exhibit a single-atom thickness, high mechanical strength, and good chemical stability [84]. Keeping in mind that the membrane thickness is inversely proportional to its permeability, graphene-

based membranes have become ideal membranes with minimum transmission resistance and maximum permeation flux because of their ultra-low thickness. However, most graphene-based materials do not have suitable natural pores, so they cannot be directly used for gas separation. Therefore, to improve the gas separation performance of graphene-based membranes, the emphasis of the research is on imparting uniformly distributed nanopores with an appropriate size and shape and high porosity in graphene sheets [85]. By designing two dumbbell-shaped porous $\gamma$-graphene monolayers containing $\gamma$-graphyne $N_2$ ($\gamma$-GYN) and $\gamma$-graphyne $H_2$ ($\gamma$-GYH), respectively, Sang et al. [86] simulated the membrane performance to separate $H_2$ from an $H_2$, $H_2O$, $CO_2$, $N_2$, CO, and $CH_4$ gas mixture. The $\gamma$-GYN membranes exhibited better selectivity and $H_2$ permeability, and they could be an ideal choice for $H_2$ purification from the gas mixture. Silva et al. [87] proved that g-$C_3N_4$ graphene-like two-dimensional nanomaterials could effectively purify $H_2$ from $CO_2$ and $CH_4$. Theoretical analyses suggested that $H_2$ permeability might be improved, by enlarging the pore area by applying 2.5 and 5% biaxial strains to the membranes, without affecting the $H_2/CO_2$ and $H_2/CH_4$ selectivity. Moreover, Wei et al. [88] used density functional theory (DFT) to study the performance of 3N-PG and 6N-PG monolayers composed of porous graphene (PG) membranes and nitrogen in separating $H_2$ from the $H_2$, CO, $N_2$, and $CH_4$ gas mixture. It was also revealed that 3N-PG monolayers and 6N-PG monolayers possessed better $H_2$ permeability than PG membranes, providing a novel membrane material for $H_2$ purification. Sun et al. [84] studied the nano-PG (NPG) membranes and found that the $H_2$ permeability reached 106 GPU, which was much higher than that of polymer membranes. At the same time, the $H_2/CH_4$ selectivity was 225, similar to that of polymer membranes. Meanwhile, NPG membranes are more cost-efficient than polymer membranes under the same separation conditions and purification requirements. Zeynali et al. [89] prepared GO nanocomposite membranes on modified alumina tubes, indicating their good $H_2$ permeability, favorable $H_2/CO_2$ and $H_2/N_2$ selectivity, and stability, accompanied with lower costs than Pd membranes. In addition, Liu et al. [90] simulated the reaction path of gas molecules through nano-graphene C216 and proved that $H_2$ could penetrate C216 membranes with a diffusion barrier of 0.65 eV. The $H_2$ selectivity to $O_2$, $N_2$, NO, $H_2O$, CO, and $CO_2$ was up to 1033, higher than that of PG and polymer membranes.

### 4.2.4. MOF Membranes

MOFs generally represent a novel class of organic–inorganic hybrid porous solid materials with regular geometric and crystal structures. They are composed of metal ions or metal ion clusters connected by organic connectors. Compared with other porous materials, MOFs have the advantages of structural variability, ultra-high porosity, uniform and adjustable apertures, adjustable inner surface properties, etc. [91]. Wang et al. [92] prepared dense and defect-free Mg-MOF-74 membranes with MgO crystal seeds and modified them with ethylenediamine. The results indicated significantly improved $H_2/CO_2$ separation performance, while the $H_2/CO_2$ selectivity increased from 10.5 to 28 at room temperature. Jin et al. [93] prepared novel CAU-10-H MOF membranes and reported their good $H_2$ permeability. The maximum separation coefficients of $H_2/CO_2$ and $H_2/H_2O$ were 11.1 and 5.67, respectively. They also found that such membranes could retain their structure and $H_2$ selectivity under long-term hydrothermal conditions, suggesting that they are suitable for $H_2$ separation in ethanol steam reforming. Liu et al. [94] synthesized a novel heterogeneous MIL-121/118 MOF membrane. The mixed $H_2/CO_2$, $H_2/CH_4$, and $H_2/N_2$ separation coefficients were 10.7, 8.9, and 7.5, respectively, at 293K and 1 bar. The average $H_2$ permeability was $7.83 \times 10^{-8}$ mol·m$^{-2}$·s$^{-1}$·Pa$^{-1}$. Meanwhile, MIL-121/118 exhibited high thermal stability and durability, showing a good application prospect.

### 4.3. Metal Hydride Separation Method

The metal hydride separation method refers to purifying $H_2$ using $H_2$ storage alloys to absorb and desorb $H_2$ reversibly. $H_2$ molecules decompose into H atoms catalyzed by $H_2$ storage alloys by lowering the temperature and increasing the pressure. Then, metal

hydrides are generated via diffusion, phase transition, combination reaction, and other processes, while impurity gases are trapped among metal particles. After the temperature is elevated and pressure is lowered, the impurity gases discharge from the metal particles, and then $H_2$ comes out from the crystal lattice. $H_2$ storage alloys can be divided into rare earth alloys, titanium alloys, zirconium alloys, and magnesium alloys based on the type of the main element. Furthermore, they can also be classified into $AB_5$-type alloys, $AB_2$-type alloys, AB-type alloys, and $A_2B$-type alloys according to the main elements' atomic ratio [95]. The performance of the $H_2$ storage alloys determines the efficiency of $H_2$ purification, so the chemical stability and tolerance of $H_2$ storage alloys can be improved and the influences of impurity gases can be reduced by modifying $H_2$ storage alloys. Dunikov et al. [96] used two kinds of $AB_5$-type alloys to separate the $H_2/CO_2$ mixture. They found that for the low-pressure $LaNi_{4.8}Mn_{0.3}Fe_{0.1}$ alloy, $H_2$ can be purified from the mixture containing 59% $H_2$ with a recovery rate of 94%, supporting the operation of PEMFCs. Yang et al. [97] carried out cyclic experiments on the $LaNi_{4.3}Al_{0.7}$ $H_2$ storage alloy in the high CO concentration environment. The $H_2$ storage capacity of this alloy slowly decreased at 363 K or higher temperatures, maintaining a relatively high kinetic rate so that it can be used for $H_2$ separation and purification in different applications. Besides this, Hanada et al. [98] studied the effects of $CO_2$ on the $H_2$ absorption performance of $AB_2$-type alloys to develop metal hydrides for $H_2$ purification and storage. The results showed that Fe and Co addition could improve the alloys' tolerance to $CO_2$, while Ni addition had the opposite effect. Zhou et al. [99] found that $MgH_2$ catalyzed by nano VTiCr easily reacted with low-pressure $H_2$, and is recycled in mixed gas. Therefore, the material showed $H_2$ separation and purification potential.

### 4.4. Cryogenic Distillation

The principle of cryogenic distillation is to separate and purify $H_2$ by utilizing the difference in the relative volatility of different components in feed gases. Compared with $CH_4$ and other light HCs, $H_2$ has relatively high volatility, such that HCs, CO, $N_2$, and other gases condense before $H_2$ with temperature reductions [100]. This process is usually used for $H_2$–HC separation. The low-temperature separation method assures a high $H_2$ recovery rate, but it is challenging to adapt the method for treating different feed gases. As such, it is necessary to remove $CO_2$, $H_2O$, and other impurities from the feed gases before the separation so as to avoid equipment blockage at a low temperature. Besides this, high costs and energy consumption accompany the requirements for gas compressors and cooling equipment in the actual operation. Although most impurities are liquefied at a low temperature, some remain in the gas phase as saturated steam, so it is difficult to directly obtain $H_2$ that meets the purity standards of fuel cell vehicles.

### 5. Conclusions and Prospects

Compared with industrial $H_2$, the purity of $H_2$ for fuel cell vehicles is not sufficient, although the requirements for the impurity content in $H_2$ are stringent. According to the existing standards, an impurity level above the limit may damage the fuel cell's performance. Thus, removing specific impurities is the focus of future research on $H_2$ purification for fuel cell vehicles. PSA is a universal method that can be applied to remove most contaminants. $H_2$-permeable membranes are often used to remove CO, $CO_2$, $N_2$, $CH_4$, $H_2O$, and other gas impurities, while $CO_2$-permeable membranes enable only $CO_2$ removal. Owing to CO, $CO_2$, and $H_2O$ sensitivity, metal hydrides can be used to remove $N_2$, Ar, and other inert gases. However, all the existing $H_2$ purification methods are limited, and it is difficult to achieve the $H_2$ impurity level standards for fuel cell vehicles by using only one separation and purification method. Since there are many different $H_2$ sources, two or even more $H_2$ purification technologies should be adopted.

**Author Contributions:** conceptualization, Z.D., C.L. and W.S.; writing—original draft preparation, C.L. and G.H.; writing—review and editing, J.Z., C.L., Y.X. and X.G.; All authors have read and agreed to the published version of the manuscript.

**Funding:** This research was funded by the National Key Research and Development Program of China (Grant No. 2019YFB1505000) and the Technology Innovation Project of China Energy Investment (Project Number: GJNY-19-136).

**Conflicts of Interest:** The authors declare no conflict of interest.

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
