# Peer review of "A Review of Hydrogen Purification Technologies for Fuel Cell Vehicles"

_catalysts, doi:10.3390/catal11030393_

Round 1

Reviewer 1 Report

The problem of hydrogen purity as a fuel in fuel cells is of key importance for the application of this technique in modern engine solutions for road transport and, in the future, also for air transport. Therefore, the presented manuscript has a chance to be noticed by a wide spectrum of readers. In my opinion, the work is presented clearly, covering a wide range of techniques used to purify hydrogen from undesirable admixtures of other gases that may adversely affect the performance of fuel cells. The overview of the techniques used for hydrogen purification, although extensive, shows that so far there is no method that meets all the requirements for hydrogen purity standards for fuel cells. The authors suggest that the PAS technique is the most promising, but I believe that the properties of metal-hydrogen systems are still underestimated. This applies to both membrane separation and the process of hydrogen formation and desorption by specific metallic alloys. The problem of material stability can be overcome by using alloys that form solid solutions rather than hydride phases. This avoids sudden volume changes in the materials that can make the alloy used brittle. A good example is a palladium-silver alloy with a silver content greater than 25%.

Although the cited literature is quite abundant, I would suggest also including the work: Separation Science and Technology Volume 35, 2000 - Issue 5 by S. Sircar & T.C. Golden.

The acronym PSA, although used many times in the text, is nowhere explained (line 116)

I would recommend this manuscript for publication with minor changes as indicated above

Reviewer 2 Report

Dear Authors,

first of all, thank you for considering Catalyts to publish your review work focused on hydrogen purification technologies for fuel cell electric vehicles applications. The reported research topic is up-to-date and interesting since no review works on this topic have been published so far. The paper is well written and structured. Valuable information for students, engineers, and researchers are reported. Before publication, I suggest you to take into consideration these following feedback to enhance the overall quality of your paper:

1) At row 59, please can you provide further details about the percentage distribution of each H2 production technology in China? For example, in US, 95% of the hydrogen is produced from natural gas. It would be interesting for readers to have a better idea of the distribution of H2 production technology in China.

2) Besides, at rows 61-62, please provide further information about the H2 production processes still under research and development?

3) To introduce the objective of the paper, please add it in a separate paragraph. It is clearer for readers.

4) Please add the plan of your paper after introducing the objective of the paper, in a separate paragraph.

5) When introducing previous works published in the literature, please use present perfect tense instead of past simplte tense.

6) From 4.1 to 4.3, it will be interesting to summarize in several tables for each subsection the previous works reported in the literature (metal membranes, polymer membranes, carbon-based membranes, etc...). It will be easier for readers.
